# Effectiveness of Surface Treatment with Amine Plasma for Improving the Biocompatibility of Maxillofacial Plates

**DOI:** 10.3390/ma12162581

**Published:** 2019-08-13

**Authors:** Yeon-Woo Jeong, Seunggon Jung, Jeong Joon Han, Hong-Ju Park, Rok Young Kim, Byung-Hoon Kim, Min-Suk Kook

**Affiliations:** 1Department of Oral and Maxillofacial Surgery, School of Dentistry, Chonnam National University, Gwangju 61186, Korea; 2KJ Meditech, Gwangju 61009, Korea; 3Department of Dental Materials, College of Dentistry, Chosun University, Gwangju 61452, Korea

**Keywords:** titanium plate, amine plasma, surface modification, hydrophilicity, biocompatibility, new bone formation

## Abstract

To date, no products have been presented for the surface treatment of metal plates used for repairing maxillofacial defects caused by trauma. Plasma surface treatment is a useful technique for chemically modifying the surfaces of biomaterials. Amine plasma-polymerization is an efficient way to prepare bioactive thin film polymers terminated with nitrogen-containing functional groups. The purpose of this study was to investigate the improvement in biocompatibility of titanium (Ti) plates treated with amine plasma-polymerization, and analyze their surfaces characteristics. To compare biocompatibility levels, in vitro test and animal study were performed using an amine plasma-polymerized Ti plate and an untreated Ti plate. After amine plasma-polymerization, the hydrophilicity of the Ti surface was remarkably improved. Biocompatibility was also improved for the Ti plates treated with amine plasma. The clinical application of this technique will not only shorten the time required for osseointegration, but will also improve the regeneration of bone.

## 1. Introduction

In the oral and maxillofacial region, various trauma-related bone defects can occur. Therefore, various efforts have been made to develop treatments for bone defect recovery and reconstruction. [1,2,3,4]. For the reconstruction of bone defects, various bone graft materials and bone substitutes are used in combination with frequent applications of metal plate fixation [5,6,7,8,9]. In the case of the maxillofacial titanium plates used in bone defect site(s), the characteristics of plate surfaces may determine initial osseointegration. If the surface roughness of a plate is non-uniform, it may lead to extended time for osseointegration or failure of osseointegration. Various surface treatments can be used to modify the physical properties of titanium surfaces, such as titanium plasma spraying, hydroxyapatite coating, sandblasted with large acid-etched grits (SLA), and resorbable blast media [10,11,12,13,14,15].

Thus far, no products have been proposed for the surface treatment of metal plates used to treat maxillofacial defects caused by trauma without dental implants. This is because the metal plates used in the maxillofacial area are fabricated for the purpose of fixing reduced bone and being removed in the future, instead of supporting significant loads, unlike dental implants [16,17]. However, if a surface-treated plate is developed for the maxillofacial area, it may be possible to promote faster recovery and regenerate higher quality bone [18,19,20,21,22,23]. Among available surface modification techniques, plasma surface modification is a useful technique for chemically modifying the surfaces of biomaterials [24,25]. In particular, amine molecules are very useful for immobilization because they form covalent bonds with bioactive substances and proteins, which can improve biocompatibility compared to plates without surface modification. Additionally, amine plasma is very inexpensive and easy to mass-produce based on its simple treatment process. Furthermore, the generation of by-products, such as ozone, is minimized during plasma generation, meaning uniform surface treatment can be achieved [26,27,28,29,30]. A lot of studies on amine plasma surface treatment for biological implants have been reported [31,32,33,34]. The plasma-polymerization of amine-based monomers is an efficient way to prepare bioactive thin film polymers terminated with nitrogen containing functional groups [35]. These surfaces are characterized by a positive surface charge in aqueous environments and thus prone to interaction, especially with hyaluronan, a negatively charged substance of the extracellular matrix, playing a key role in the initial adhesion of osteoblasts to artificial surfaces [35]. However, no studies have been reported of the amine plasma-polymerization technique applied to titanium plate surface coating. 

The purpose of this study was to investigate the improvement in biocompatibility of titanium plates treated with amine plasma through in vitro/in vivo evaluation and analyze their surfaces. Furthermore, after implanting different plates (plates treated with amine plasma and untreated plates) into leporine crania with bone defects, bone regenerative capacity was measured on the bone-side surfaces.

## 2. Materials and Experimental Methods

### 2.1. Materials

In this study, two kind of titanium (Ti) disks (diameter = 16 mm, thickness = 3 mm) and Ti plates (length = 10.85 mm, thickness = 1 mm, hole diameter = 2 mm) were used as samples for cell response and animal study, respectively. Samples were supplied by KJ Meditech Company, Gwangju in Korea. One is an untreated Ti surface (control group) and the other is sandblasted with an SLA-treated Ti surface (experimental group). In general, commercially Ti maxillofacial plate was treated with SLA for improving the osseointegration. Allylamine monomer was purchased from Sigma-Aldrich. All other chemicals and bioreagents were used without further purification. Figure 1 shows the samples used in this study.

### 2.2. Amine Plasma-Polymerization on Sandblasted with Large Acid-Etched Grits (SLA)-Treated Ti Surface 

The amine plasma-polymerization of SLA treated Ti disks and plates were carried out by depositing an ultra-thin polymeric film containing amine groups through plasma-polymerization of allylamine monomer. The equipment used for amine plasma-polymerization was a low-pressure radio frequency (RF) glow discharge plasma deposition system (model UM1000134-001A Miniplasma-station, PLASMART, Daejeon, Korea). Amine plasma-polymerization was performed under the process conditions listed in Table 1. Here, preconditioning process was performed to activate and clean on Ti surface, and post-processing was performed to enhance the degree of polymerization of the polyallylamine ultra-thin film.

### 2.3. Tiatanium Surface Characterization after Amine Plasma-Polymerization

#### 2.3.1. Hydrophilicity Evaluation

Following amine plasma-polymerization, the contact angles of the modified titanium surface were measured to evaluate its hydrophilicity. The contact angle was examined by the sessile drop method using a goniometer (GS, Surface Tech Co. Ltd., Gwangju, Korea). A water droplet (7 μL) was dropped using a syringe mounted vertical to the sample surface. The measurement was repeated five times for each experimental and control group.

#### 2.3.2. Surface Analysis of Polyallylamine Thin Film Deposited by Amine Plasma-Polymerization 

Following amine plasma-polymerization, Fourier transform infrared spectrometry (FT-IR, Perkin Elmer, Spectrum Two^TM^ FT-IR Spectrometer, Buckinghamshire, UK) was used to analyze the polymer thin film on the titanium surface [36,37]. The resolution was 4 cm^−1^ and eight scans were accumulated.

#### 2.3.3. Evaluation of Amine Concentration 

To determine the amine concentration on the amine plasma-polymerized titanium disk surface, a methyl orange (MO) assay was used. The MO colormetric measurement can be used to determine the number of amino groups on polymer surfaces. In brief, amino functionalized Ti samples were treated with the acidic methyl orange solution for 1 h. The absorbance of the methyl orange solution desorbed in a desorbing agent was measured at 465 nm [36]. The −NH_2_ content was obtained from a calibration plot of the optical density versus dye concentration with a 1:1 ratio assumed between the dye and amino groups [21]. 

#### 2.3.4. Surface Chemistry Analysis 

The amine plasma-polymerized Ti surfaces were also characterized by X-ray photoelectron spectroscopy (XPS, VG Multilab 2000, Thermo VG Scientific, London, UK) to obtain the elemental composition and chemical state. For each specimen, a compositional survey scan was acquired using a pass energy of 50 eV and the core level spectra were acquired with a pass energy of 20 eV and a step size of 0.5 eV. All surface compositions reported are expressed in atomic percent (at.%) values. 

### 2.4. In Vitro Evaluation of Titanium Plates following Amine Plasma Treatment

#### 2.4.1. Culture of Cell Lines 

The cell line used in this experiment consisted of osteoblastic cells (MC3T3-E1) from a mouse skull. These cells were purchased from ATCC. 

Cell cultures were supplemented with growth factors (10% (w/v)) fetal bovine serum (FBS, PAA Laboratoris, Inc. A15-751) and antibiotics (100 units/mili-liter (mL) penicillin and 100 μg/mL streptomycin) in an α-MEM medium (alpha minimum essential medium with ribonucleosides, deoxyribonucleosides, 2 mM L-glutamine, and 1 mM sodium pyruvate, but without ascorbic acid/GIBCO, Custom Product, Catalog NO. A1049001, Thermo Fisher Scientific, Waltham, MA, USA). The cells were cultivated in a CO_2_ incubator (Sanyo Electric, Osaka, Japan) maintained at 100% humidity and 37 °C with 5% CO_2_. Subcultured third generation cell lines were used for cell proliferation assay at 80% confluency every three days.

#### 2.4.2. Evaluation of the Osteoblast Proliferation

The proliferation rate of osteoblastic cell lines was evaluated using a MTT (3-(4,5-Dimethyl-2-thiazolyl)-2,5-diphenyl-2H-tetrazolium bromide)) assay based on the following process. After removing all excess α-MEM medium, the cultured MC3T3-E1 cells were washed with phosphate buffered saline (PBS, Sigma-Aldrich, St. Louis, MO, USA) and separated from the culture dish by adding trypsin/ ethylenediaminetetraacetic acid (EDTA, Sigma-Aldrich, St. Louis, MO, USA). FBS was then added to the separated cells to halt any reactions and the cells were collected using a centrifuge. After the cells were suspended by adding medium, they were seeded at 5 × 10^4^ cells/well in a 12-well plate containing the prepared sample. On the first, third, and fifth days, 100 μL of MTT (M2128, Sigma-Aldrich, St. Louis, MO, USA) reagent was added to each well and it was confirmed that blue-violet crystals were formed. After 4 h, 1,000 μL/well of dimethyl sulfoxide (35535-0350, Junsei Chemical Co., Ltd., Tokyo, Japan) was added, followed by incubation at room temperature for 30 min. To measure absorbance, 200 μL of each reaction mixture was dispensed into a 96-well plate and the absorbance was measured at 540 nm using an EPOCH device (BioTek Instruments, Inc., Winooski, VT, USA).

#### 2.4.3. Observations of Cytoskeletons and Nuclei of the Osteoblast

MC3T3-E1 cells were seeded at a concentration of 2 × 10⁵ cells/mL in a 12-well plate for each sample. After 48 h of cell seeding, the cells were fixed with a 4% paraformaldehyde (Electron Microscopy Sciences 15714) solution and washed with PBS for 5 min. The cells were then treated with 0.1% Triton X-100 (T1020, Bio-Sesang, Seongnam, Korea) and 1% bovine serum albumin(BSA, SIGMA A9647, Sigma-Aldrich, St. Louis, MO, USA) for 5 min. A Rhodamine-Phalloidin (R415, Thermo Fisher Scientific, Waltham, MA, USA) reagent was applied for 15 min to stain the cells. After washing the cells with PBS three times for 5 min each, the samples were then attached to cover glasses using a fluorescence mounting media (VECTOR H-1200, Vector Laboratories, Burlingame, CA, USA) for nuclei staining and stored at 4 °C for further observation using a fluorescence microscope [38].

#### 2.4.4. Evaluation of Osteoblasts Differentiation

The differentiation of osteoblasts was confirmed by measuring alkaline phosphatase (ALP) activities. The cells were cultured on an untreated titanium plate and amine plasma-treated titanium plate at a concentration of 2 × 10^5^ cells/mL. The original medium was replaced with a mixed medium containing α-MEM (GIBCO), 50 μg/mL of ascorbic acid (JUNSEI), and 10 mM of β-glycerophosphate (Sigma-Aldrich) the following day. To evaluate the level of differentiation, p-nitrophenol (p-NP) release from p-NP phosphate was measured after 5 d. The medium containing MC3T3-E1 cells was washed twice with Dulbecco’s PBS (Welgene) and again with a 0.9% NaCl solution containing 0.2% of Triton X-100 (Sigma-Aldrich) for 10 min. The cells were sonicated for 1 min at 65 W on ice using a Vibra-Cell instrument (SONICS, Newtown, CT, USA). The washed lysate was centrifuged at 4 °C and 2500 × *g* for 10 min. The supernatant was cultivated using a p-NP phosphate solution for 30 min at 37 °C. The reaction was halted by adding 600 μL of 1.2-N NaOH. ALP activities were confirmed by measuring absorption at 405 nm and normal protein concentration using an enzyme-linked immunosorbent assay (ELISA) reader (BioTek Instruments, Inc. Winooski, VT, USA). Protein concentrations were determined using Bradford protein quantification (Bio-Rad Protein Assay, Bio-Rad Laboratories, Inc., Hercules, CA, USA) and the results were expressed as μmole of p-NP/min/μg protein.

#### 2.4.5. Evaluation of Bone Calcification

Bone calcification was measured using Alizarin Red S staining. Differentiated cells were washed twice using a Dulbecco’s PBS solution and fixed with a 10% formalin solution for 15 min. Following fixation, the cells were washed twice with distilled water. Shortly thereafter, an Alizarin Red S solution was mixed with distilled water and an NH_4_OH (ammonium hydroxide) solution was added to titrate the pH level to 4.1. The treated Alizarin Red S solution was added to the fixed cell mixtures to cover all cells and was covered with foil for approximately 45 min. The Alizarin bonded to the Ca^2+^ in the cell matrix exhibited a reddish-orange color.

### 2.5. Animal Study

Domestic male rabbits weighing 2 kg were used as the experimental subjects. Experiments were conducted on a total of 20 rabbits. Two-hole straight plates with no treatment and two-hole straight plates with amine plasma treatment were tested.

The study protocol was approved by the Institutional Review Board at Chonnam National University Hospital. The authorization number is CNU IACUC-H-2018-32. All surgical procedures were performed under sterile conditions. The experimental procedure was as follows. The day before the experiment, sterilization using an autoclave was performed on all surgical instruments used during animal experimentation (121 °C. 15 psi, 15 min). The rabbits were weighed and 3 mL of Zolethyl (Zolethyl 50, Virbac Co, Carros, France) per kg bodyweight was injected intramuscularly into the thigh to induce sedation and anesthesia. The rabbits were placed in the prone position and shaved from the bregma to the supraorbital margin. Povidone iodine, chlorhexidine gluconate, and isopropanol were then used to disinfect the surgical site. Infiltration anesthesia was administered with lidocaine containing 1 ppm of epinephrine at the surgical site. The cranium was exposed after incision and dissection of subcutaneous tissue and the periosteum using a No. 15 blade. After exposing the cranium, a round bur was used under irrigation to form a shallow groove in which a small metal plate could be placed. Groove formation also causes moderate bleeding to promote osteogenesis. Each rabbit received one untreated titanium plate and one treated with amine plasma. These two-hole miniplates were fixed with screws for both test groups. The periosteum was sutured using 3-0 vicryl, followed by the skeletal muscle, subcutaneous tissue, and skin using 3-0 vicryl, 5-0 vicryl, and 6-0 nylon, respectively. The surgical sites were disinfected using povidone iodine, chlorhexidine gluconate, and isopropanol, mirroring the pre-operation procedure. Finally, Zenocef^®^ (first-generation cephalosporin antibiotic) and Keromin^®^ (non-steroidal anti-inflammatory drug) were injected. The systemic conditions of the rabbits, as well as the healing and infection conditions at the surgical sites, were monitored daily and the sites were disinfected. After four weeks, the rabbits were euthanized using a KCl solution and bone fragments with a margin of 5 mm around the small metal plates were extracted using an osteotome. Figure 2 shows the extracted Ti mini plate samples used in animal study. 

### 2.6. Statistical Analysis

In vitro data were statistically analyzed using a Kolmogorov–Smirnov (KS) test to check for normality. A student’s t-test was then used based on the normal distribution identified by the KS test. A threshold of p ≤ 0.05 was considered to be significant (*****) and a threshold of p ≤ 0.01 was considered to be highly significant (******).

Histomorphometric analysis of newly formed bone was carried out using an Image J^®^ Version 1.52 device (National Institutes of Health, Wayne Rasband, Bethesda, MD, USA). After tracing the new bone growth, the pixels in the traced area were counted. The mean results were compared through a paired t-test using the SPSS 23.0 TM software (IBM, Armonk, NY, USA). The differences were considered to be statistically significant at a threshold of p < 0.05.

## 3. Results and Discussion

### 3.1. Changes of Ti Surface Characterizations after Amine Plasma-Polymerization

#### 3.1.1. Determination of Amine Concentrations 

Figure 3 shows the optical density of methyl orange solution desorbed and amine concentrations as a function of plasma RF powers. As the shown in Figure 3, amine concentrations on the Ti surface were 41 nmol/cm^2^ at 30 W, 64 nmol/cm^2^ at 50 W, and 73 nmol/cm^2^ at 70 W, indicating that as power increases, the amine concentration on the surface also increases. The highest degree of NH_2_ density on the surface is 73 nmol/cm^2^, was detected at the higher RF power 70 W. The hydrophilicity is influenced by the –NH_2_ grafted density on the modified Ti surface, so we selected the optimized condition for the plasma-polymerization process (RF power 70 W). Therefore, we treated all plasma-polymerized Ti samples used in the in vitro and animal studies with this condition.

#### 3.1.2. Identification of Polyallylamine Ultra-Thin Film Deposited on the Ti Surface

To confirm the presence of polyallylamine thin film after the amine plasma-polymerized Ti surface, FT-IR analysis was performed. Figure 4 shows the FT-IR spectra of the SLA-treated Ti surface and amine plasma-polymerized Ti surfaces. As the shown in Figure 4, the FT-IR spectrum of the SLA-treated Ti surface did not shown the amines absorption bands, on the other hand, amine plasma-polymerized Ti surface clearly shows several absorption bands characteristic of the presence of primary (1600 cm^−1^) and/or secondary amines (3380 cm^−1^) groups. Based on this result, the presence of a polyallylamine ultra-thin film on the surface of amine plasma-polymerized titanium disk was confirmed. Therefore, it is clear that the plasma surface modification of Ti surface through amine plasma-polymerization was successful.

#### 3.1.3. Surface Chemistry Analysis

To examine the surface chemistry of the Ti sample after amine plasma-polymerization, X-ray photoelectron spectroscopy (XPS, VG Multilab 2000, Thermo VG Scientific, London, UK) analysis was performed. The surface compositions determined from the wide scan XPS spectra are shown in Figure 5. The peaks at 284.8 and 531.2 eV were present in untreated Ti samples and assigned to C 1s and O 1s, respectively. The peak at 399.9 eV, corresponding to N 1s, was detected in amine plasma-polymerized Ti surface only. From the XPS analysis results, we confirmed that amine functional group containing polyallylamine thin film was successfully deposited on Ti surface thorough plasma-polymerization process. 

#### 3.1.4. Hydrophilicity

The hydrophilicity was determined by measuring the water contact angle (WCA) on the plasma-polymerized Ti surface. Figure 6 shows the contact angle values and representative images of the corresponding water droplets. It can be seen that there are obvious differences in the wetting properties before and after plasma-polymerization. The contact angle of the untreated titanium surface was 77.18°, representing strong hydrophobicity. In contrast, the contact angle of the surface treated with amine plasma was 28.14°, representing strong hydrophilicity. These results confirm that changes in surface chemical compositions can affect the wettability of the materials, which is a useful parameter that has often been strongly correlated with the cell-biomaterial interfacial interactions [38].

### 3.2. In Vitro Evaluations 

#### 3.2.1. Proliferation Rate of Osteoblastic Cell Lines

Cell proliferation was measured using an MTT assay. Figure 7 presents the cell proliferation rates after one, three, and five days of cell culturing. As a control, the cell proliferation rate on the surface of the untreated titanium plate was defined as 100%. The Ti samples with amine plasma-polymerization showed higher MC3T3-E1 proliferation than untreated Ti samples. All the experimental groups for MTT assay showed statistically significant differences (p < 0.01, in comparison to untreated Ti sample).

#### 3.2.2. Observations of Cytoskeletons and Nuclei of the Osteoblastic Cell Lines

Rhodamine-phalloidin and 4′,6-diamidino-2-phenylindole (DAPI) fluorescence staining were performed to observe the cell morphologies attached to the titanium plates (Figure 8). Four h after cell seeding, fluorescence microscopy revealed that the number of cells was greater on the titanium surface treated with amine plasma than on the untreated titanium surface. Additionally, cytoskeletons were spread widely on the titanium surface treated with amine plasma, confirming the enhanced biocompatibility of the treated surface. 

#### 3.2.3. Differentiation of Osteoblastic Cell Lines 

ALP activities were measured to analyze the differentiation of cells attached to the titanium plates. After seven days of osteoblast cultivation, ALP activities were measured on the untreated titanium surface and the titanium surface treated with amine plasma. It was apparent that the level of differentiation increased significantly in the sample treated with amine plasma (Figure 9). All the experimental groups for ALP activity showed statistically significant differences (p < 0.01, in comparison to untreated Ti sample).

#### 3.2.4. Bone Calcification

Alizarin Red S staining was performed to examine the calcification of bone generated on the titanium plates. Osteoblasts were cultivated on both the untreated titanium surface and the titanium surface treated with amine plasma for 14 d. As the shown in Figure 10, amine plasma-polymerized Ti disk shows the strong reddish color compare to the untreated Ti disk. Therefore, the sample treated with amine plasma exhibited excellent bone calcification. 

### 3.3. Animal Study

#### 3.3.1. Clinical Evaluation

On the day of euthanizing the rabbits, the following visual observations were made. On the soft-tissue-side surfaces of the Ti plates in the experimental group where polishing was performed during dissection, the dissection was relatively clean compared to the control group (Figure 11). Additionally, the amount of periosteum and inflammation remaining around the metal plate was relatively small and the smooth surface of the bone could be observed.

#### 3.3.2. Histological Findings

Figure 12 shows the hematoxylin and eosin (H&E) staining of the tissue around the plates after four weeks for the untreated two-hole straight plates and amine plasma-polymerized plates. The histological findings from photographs of the collected samples in Figure 12 are discussed below. Compared to the control group, dissection was cleaner on the soft-tissue-side surfaces of the experimental group, which showed little or no inflammatory tissue. The experimental group treated with amine plasma had closer contact between the small metal plate and newly formed bone layer compared to the control group. Additionally, there was very little fatty tissue intercalating with the soft tissue.

#### 3.3.3. Histomorphometirc Findings

The ratio of the length of the newly formed bone to the length of the entire Ti plate on the bone-side surface was greater for the plasma-treated group compared to the control group (Figure 13).

The mean values of the lengths of the six bone-contacting tissues for the control and experimental groups are listed in Table 2.

#### 3.3.4. Micro-Computed Tomography (Micro-CT) Findings

Figure 14 shows the micro-computed tomography (micro-CT) images of the untreated two-hole straight plates and amine plasma-polymerized plates after four weeks. As the shown in Figure 14, micro-CT data revealed that the amount of newly formed bone on the bone-side surfaces of the experimental group was significantly higher than that on the corresponding surfaces of the control group. Additionally, the distance between the metal plate and layer of newly formed bone was relatively small for the experimental group. These results match the results of microscopic evaluation.

Our experimental results confirmed that titanium treated with amine plasma is biocompatible.

As reported by Kristen R. Kull et al., the hydrophilicity of amine plasma treated titanium plate was increased compared to that of untreated titanium plate [39]. This may lead to a significant difference in terms of biocompatibility when considering the hydrophilic environment of the bone surface. In addition, FT-IR analysis showed that the amine group was actually present on the surface of the titanium plate, and the concentration of the amine group on the surface of the titanium plate also increased with an increase in the intensity of the polymer [28]. Amine plasma treatment is not so complicated, so it is possible to fabricate a titanium plate with higher biocompatibility by polymerizing at stronger intensity in clinical practice. Animal studies artificially induced some bleeding to further activate bone regeneration. In order to minimize the differences between individuals, a titanium plate treated with amine plasma and a titanium plate treated with no treatment were plated for each individual. The proliferation rate and degree of differentiation of osteoblasts measured in this same environment showed a great difference in two types of titanium plates. As time passed, the difference in the proliferation rate of the osteoblasts between the titanium plate treated with no treatment and the titanium plate treated with the amine plasma was found to be large, and the difference was more than twice as high as on the fifth day of culture. Fluorescence microscopy showed that the number of nuclei in the titanium plate treated with amine plasma was much higher than that of the untreated titanium plate and that the cytoskeleton was more developed [40]. These results accurately reflect the results obtained from the specimens collected from animal studies. Histological findings showed that the titanium plate treated with amine plasma showed more uniformly formed new bone than the untreated titanium plate. And this is the same in micro CT. These results suggest that amine plasma treatment on titanium plate may increase biocompatibility and osteogenesis. 

However, in the case of sprayed plasma-coated titanium, chemical changes on the surface may cause changes in physical properties. Therefore, additional studies on the tensile load, compressive load, bending load, torsion load, and amount of deformation of titanium plates treated with amine plasma are required [41].

Additionally, titanium powder is very reactive, making it difficult to reproduce accurate samples using sprayed plasma coating. Therefore, it is necessary to study safe methods for the stable and accurate surface treatment of titanium with plasma [42,43,44]. 

Nutritional supply may also be important for the formation of new bone. Therefore, if nutrient materials such as polyglycolic acid, one of the bioreactors, are provided with amine treatment, the regeneration of bone will be improved [45].

## 4. Conclusions

In this study, we found that titanium surface treatment through amine plasma-polymerization was successfully performed on maxillofacial titanium plates used in clinical surgery. We characterized the amine plasma-treated Ti surface, and investigated the effectiveness of amine plasma-polymerization in improving the biological activities in both in vitro and in vivo tests. After plasma surface treatment, polyallylamine ultra-thin film was deposited on Ti surface and hydrophilicity of Ti surface was improved. Amine groups contained polyallylamine on the Ti surface positively influenced osteoblast cell behaviors, such as proliferation, and differentiation. In addition, results of animal test showed the improvement of new bone formation. Therefore, the clinical application of this technique can not only shorten the time required for osseointegration, but also improve the regeneration of bone. Based on the results, the superior bone regenerative capacity of plates treated with amine plasma is expected to increase their therapeutic effects in clinical practice.

## Figures and Tables

**Figure 1 materials-12-02581-f001:**
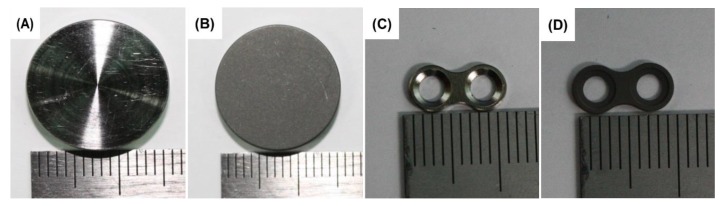
A Ti disk with no treatment (**A**) and sandblasted with large acid-etched grits (SLA) treated Ti disk (**B**). Ti disks were used as sample for surface analysis and in vitro evaluation. Untreated Ti mini plate (**C**) and SLA treated Ti mini plate (**D**). Ti mini plates were used as sample for animal study.

**Figure 2 materials-12-02581-f002:**
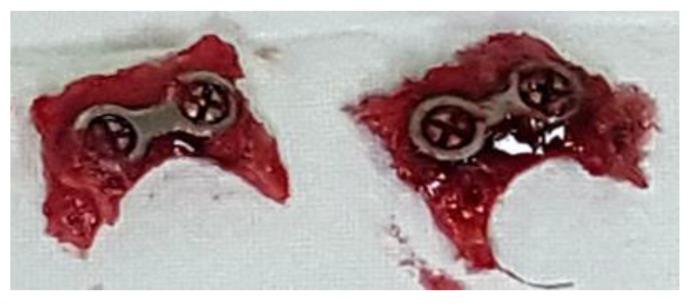
Samples extracted four weeks after the operations.

**Figure 3 materials-12-02581-f003:**
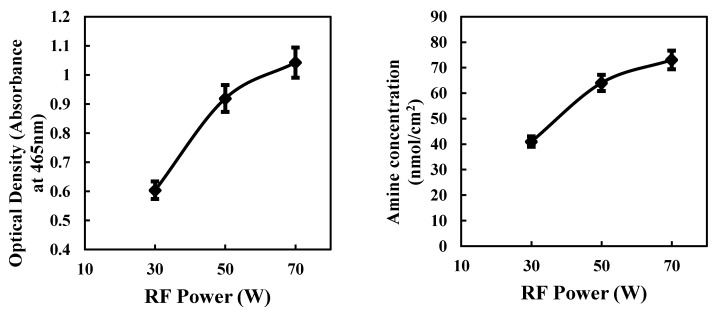
Optical density values of methyl orange solution (**left**) and amine concentrations on the titanium surface (**right**) with increasing radio frequency (RF) plasma powers.

**Figure 4 materials-12-02581-f004:**
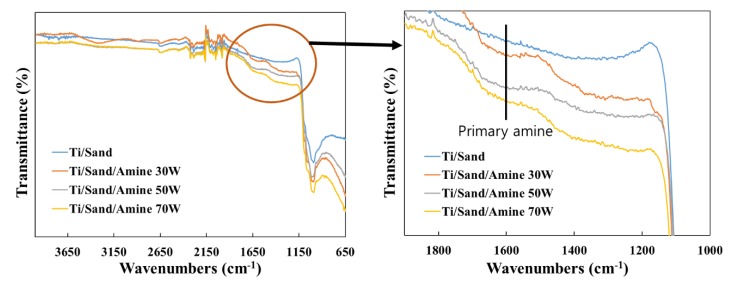
Fourier transform infrared (FT-IR) spectra of the SLA-treated Ti surface and amine plasma-polymerized Ti surfaces under three different plasma powers.

**Figure 5 materials-12-02581-f005:**
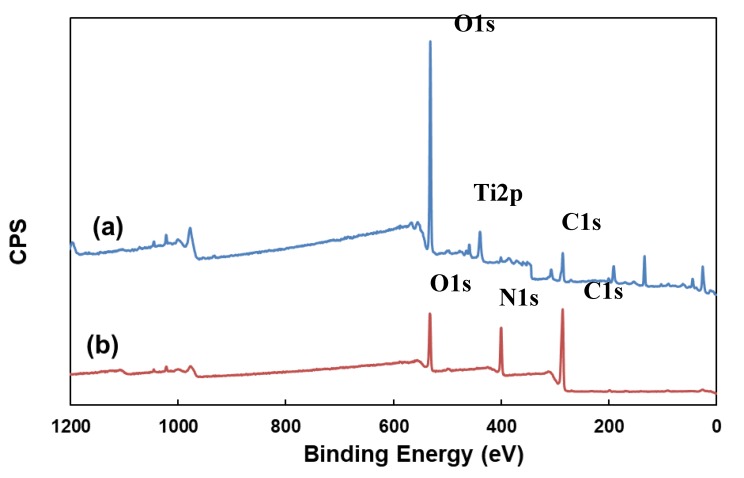
X-ray photoelectron spectroscopy (XPS) spectra of (**a**) untreated Ti surface and (**b**) amine plasma-polymerized Ti surface.

**Figure 6 materials-12-02581-f006:**
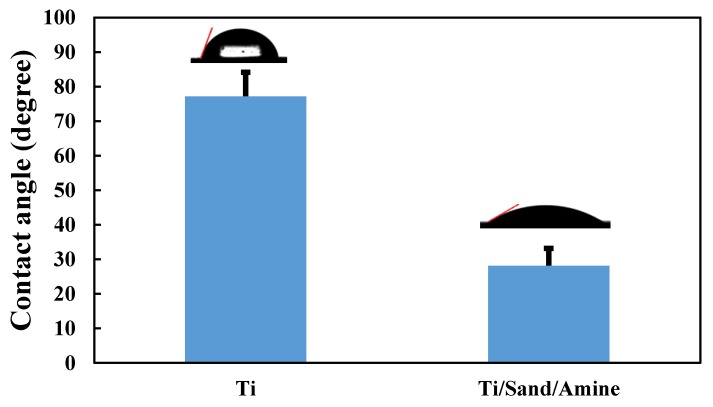
Contact angles of the untreated titanium surface and titanium surface treated with amine plasma-polymerization.

**Figure 7 materials-12-02581-f007:**
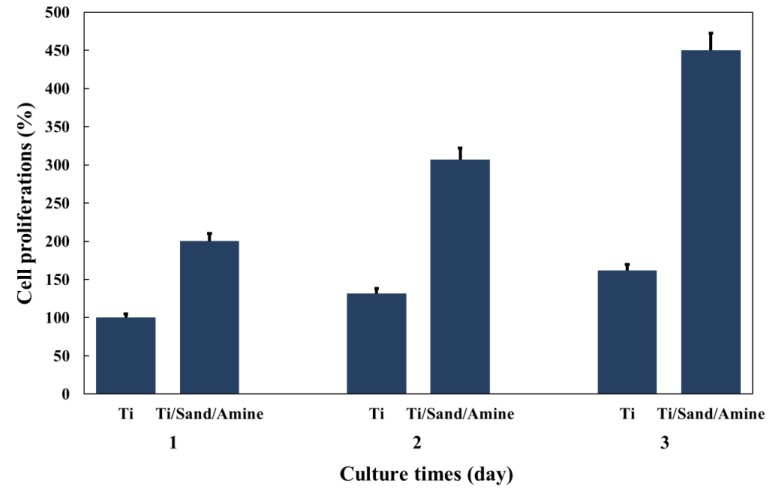
Untreated titanium surface and titanium surfaces treated with amine plasma. Proliferation rates of cultured osteoblasts on the surfaces after one, three, and five days.

**Figure 8 materials-12-02581-f008:**
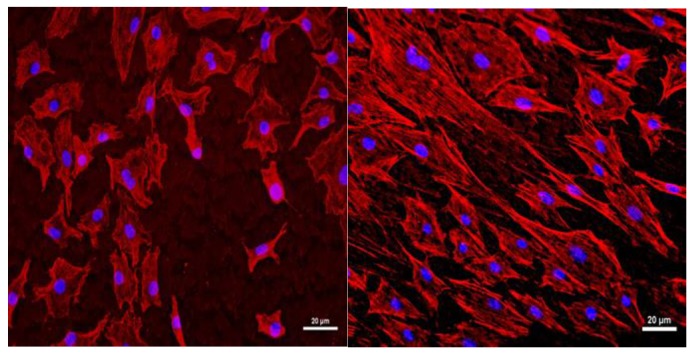
Cell morphologies on the surfaces of plates: untreated titanium plate (**left**) and titanium plate treated with amine plasma (**right**). Nuclei color is blue.

**Figure 9 materials-12-02581-f009:**
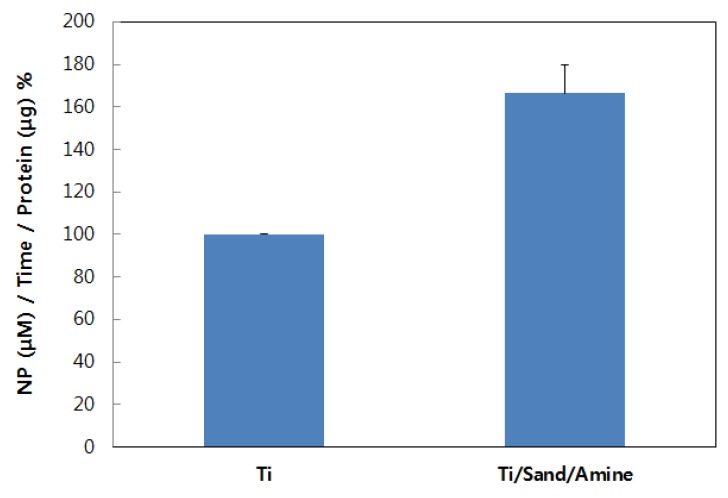
Alkaline phosphatase (ALP) activities of osteoblasts cultured for seven days on the untreated titanium surface and the titanium surface treated with amine plasma.

**Figure 10 materials-12-02581-f010:**
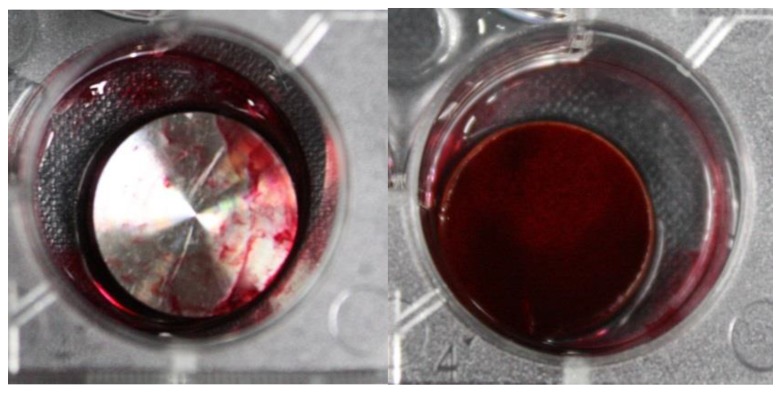
Levels of calcification on the surfaces of plates: bone calcification on the untreated titanium plate (**left**) and on the titanium plate treated with amine plasma (**right**).

**Figure 11 materials-12-02581-f011:**
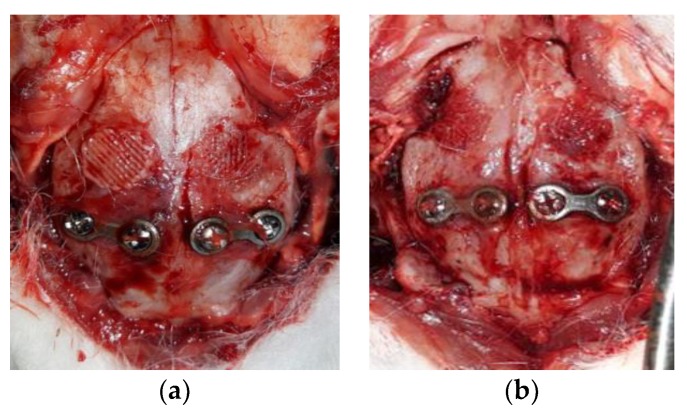
Four weeks after the initial operation, the untreated two-hole straight plate (**a**) and two-hole straight plate treated with amine plasma (**b**) were dissected. The periosteum and surrounding tissue remained around the untreated two-hole straight plate. In contrast, a relatively clear separation of tissue around the two-hole straight plate treated with plasma can be observed.

**Figure 12 materials-12-02581-f012:**
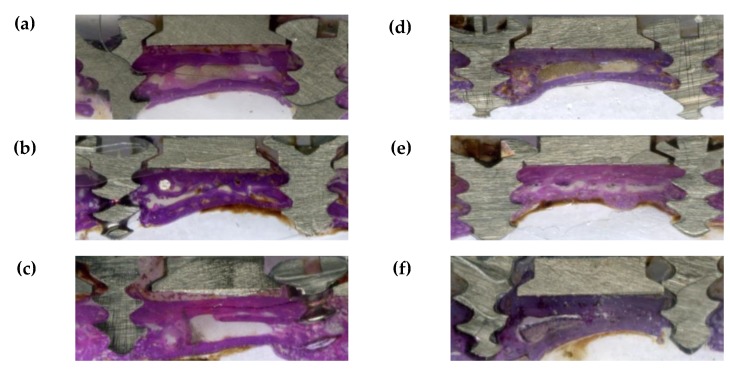
(**a**–**c**) Hematoxylin and eosin (H&E) staining of the tissue around the plates after four weeks for the untreated two-hole straight plates. A foreign body can be observed around the plates and the bone is tightly connected in some sections. Additionally, inflammatory tissue can be observed on the bone-side surfaces. (**d**–**f**) H&E staining of the tissue around the plates after four weeks for the two-hole straight plates treated with amine plasma. A small amount of foreign material can be observed around the plate and the newly formed bone on the bone-side surface is more tightly connected to the plates compared to the untreated two-hole straight plate group.

**Figure 13 materials-12-02581-f013:**
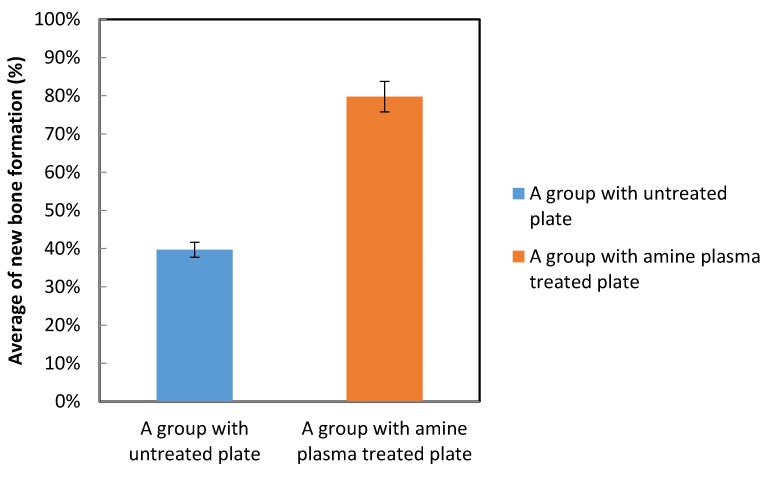
Comparison of the average percentage of new bone formation for each specimen in the animal experiment.

**Figure 14 materials-12-02581-f014:**
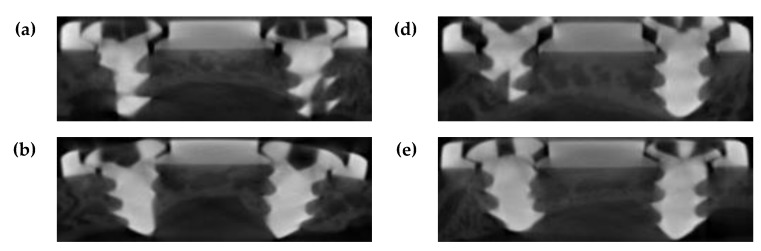
(**a**–**c**) Micro-computed tomography (micro-CT0 images of the untreated two-hole straight plates after four weeks. One can see that weak bone formed on the bone-side surface of the metal plate. (**d**–**f**) Micro-CT images of the two-hole straight plates treated with amine plasma after four weeks. One can see that dense bone formed on the bone-side surface of the titanium plate.

**Table 1 materials-12-02581-t001:** Process conditions for amine plasma-polymerization.

Treatment Conditions	Pre-Treatment	Polymerization	Post-Treatment
Plasma gas Radio Frequency discharge power Deposition time	Argon 100 W 300 s	Monomer-Allylamine 30 W, 50 W, 70 W 300 s	Argon 30 W 60 s

**Table 2 materials-12-02581-t002:** Average lengths of bone-to-plate contact areas (the distance between the two screws is 15 mm).

-	Average Length of Bone-to-Plate Contact Area (mm)	Average Percentage of New Bone Formation (%)
Group with Untreated Plates	6.06	39.70
Group with Amine Plasma-Treated Plates	12.18	79.76

(p = 0.005).

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
