# Peer review of "Effectiveness of Surface Treatment with Amine Plasma for Improving the Biocompatibility of Maxillofacial Plates"

_materials, 2019, doi:10.3390/ma12162581_

Round 1

Reviewer 1 Report

The present publication investigates a protocol of treating Ti plates used in maxillofacial transplants that augments the biocompatibility of the plates. The paper also purports to investigate the surface modifications occurring on the Ti surface after treatment with amine plasma. It is concluded that this treatment will shorten the time required for osseointegration, and also improve the regeneration of bone.

The following are general comments on the quality of the manuscript:

Minor typos and grammatical errors abound in the paper. A thorough revision is warranted prior to resubmission. The following figures need to recreated: Fig 5 - show the contact angles on the Figure itself. Fig 6 - clear, legible markings for axes, correct axes labelling (cm-1, not cm-1), overall increase in figure size, lines to be redrawn using bold colors and thicker line sizes. Figure 7 - remove the grey background, axes labelling to be done in consistent fonts and sizes. Error bars? Replicate experiments? Figure 8 - better, larger graph needed. What is the error associated with each bar? Figure 14 - remove gridlines, move the legend outside the body of the graph.

There are no citation numbers in the body of the paper, although a reference list is supplied. What’s with that?

The following issues need to be addressed prior to resubmission:

The paper purports to analyze the surfaces of Ti plates treated with plasma. In reality, beyond surface IR, very little analysis is done. No speculation about the nature of Ti surfaces (are they oxidized? If so, to what state?) or the amine species deposited on the Ti surface are made in the paper. This should be clearly addressed.  No mention is made of other studies that have used amine plasma treatment in biological implants. Is this a pioneering study? Are there other kinds of plasma used for this purpose? Why is this method an improvement over existing ones? These issues are poorly explained in the introduction. The amine monomer used was allylamine. Relevant literature states: “Allylamine is a specific cardiac toxicant that causes aortic, valvular and myocardial lesions in many species. Myocardial necrosis can be observed 24 hr after a single dose. Acute toxicity is believed to involve metabolism of allylamine to highly reactive acrolein (2-propenal). Allylamine has been shown to bind to mitochondria from aorta and heart, suggesting that the subcellular site of injury is at or near the mitochondrion. Mice exposed to allylamine died during the course of a 10 minute inhalation. Allylamine was extremely irritating when tested on rabbit eyes.” It is understood that allylamine doesn’t survive in plasma, but what are the actual amine species deposited on the plates? Are they irritants? Why not choose a less toxic amine species for the plasma? The paper states “To determine the amine concentration on the titanium plate, a reaction was carried out using a methyl orange solution. The absorbance of the methyl orange solution desorbed in a desorbing agent was measured at 465 nm.” No data for this section, however, is present in the main body of the paper. No SI section was present for this manuscript, either. Please include the spectrophotometric data or remove the section. Maybe a citation here would have helped to make things clearer, but as noted before, no citation numbers were given anywhere in the paper. Were the MTT assays for the cell proliferation studies done in replicate? What are the associated error bars? The main conclusion of the paper - that it is the amine treatment that makes the Ti plates more biocompatible - is debatable unless it is confirmed by some kind of X-ray spectroscopy that no change in the oxidation states of Ti on the plate surfaces is taking place. Furthermore, unless an actual chemical species with a reasonable structure is proposed as the capping molecule for the plate surface that confers enhanced biocompatibility onto it, I would hesitate to accept the author’s conclusions regarding the suitability of this protocol for actual usage. A control experiment set of Ti plates treated with a non-amine plasma might also throw some light on the utility of this specific way of enhancing biocompatibility of Ti plates.

I’d be happy to take a look at this manuscript again once these concerns have been addressed.

Reviewer 2 Report

Thank you for the interesting paper.

There are few concerns I have and they have to be answered.

When you mention “After the cells were suspended by adding additional medium, they were seeded at 5 × 104 cells/well in a 12-well plate containing the prepared sample.” did you use non-adherent plate? I don’t understand what’s the point of adding the induction media when your original cells are osteoblast. Evaluation of Differentiation does not make any sense, Figure 8 can’t be shown without standard deviation. Hopefully repetitions per group are at least 3. Figure 9 titanium plate treated with amine plasma on the right is very interesting. Cells are lined up in a specific direction, can you explain this a little more?

Round 2

Reviewer 1 Report

The revisions have dramatically enhanced the quality of the paper, and other than correcting some typos and grammatical errors that are still present, I recommend accepting the paper for publication.